# DYNAMIC CONTEXT ADAPTERS: EFFICIENTLY INFUSING HISTORY INTO VISION-AND-LANGUAGE MODELS

## ABSTRACT

Historical context integration presents a fundamental challenge for Vision-Language Models (VLMs) in sequential decision-making tasks. Current VLMs process visual inputs independently, which creates critical limitations for downstream applications that require temporal understanding. Direct incorporation of historical frames into Transformer inputs produces quadratic attention complexity and excessive memory consumption. Existing approaches suffer from significant drawbacks: computational inflation or substantial information loss through temporal compression. To address these challenges, we introduce Dynamic Context Adapter (DCA), a novel context injection approach for pretrained VLMs. Our method employs fixed-size, dynamically compressed memory to preserve historical semantics without frame concatenation. DCA bridges static VLMs and recurrent policies and enables memory capabilities in pretrained models while maintaining computational efficiency. DCA achieves over 25% reduction in attention FLOPs and 13% memory savings while improving performance on long-horizon tasks.

## 1 INTRODUCTION

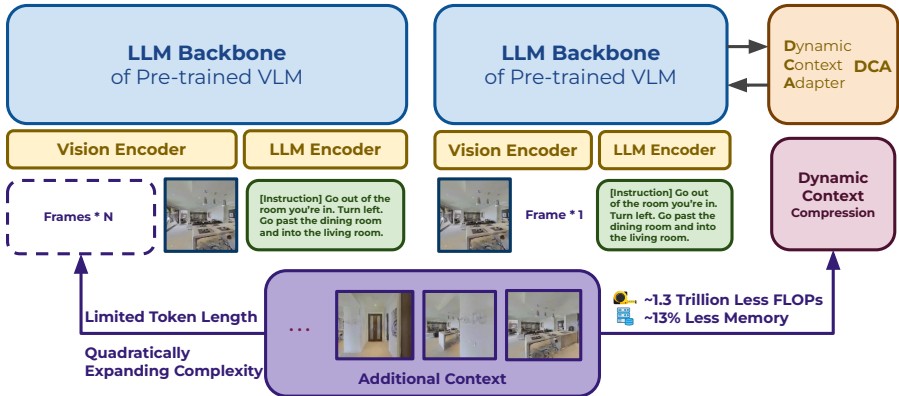

Figure 1: **Context Concatenation vs. Dynamic Context Adaptation. (Left)** Traditional concatenation appends historical frames to current input, producing quadratic computational complexity and token length constraints. **(Right)** Our proposed DCA method decouples historical context from the LLM backbone through lightweight adapters. The Dynamic Context Compression module processes historical frames and distributes compressed representations across multiple VLM layers, maintaining constant input length while achieving about $\sim 1.3$ trillion fewer FLOPs and 13% memory reduction.

Sequential decision-making in partially observable environments demands agents to integrate rich historical context across extended temporal horizons. In Vision-and-Language Navigation (VLN), agents must synthesize information from multiple past observations to navigate complex multi-room environments where current visual input alone provides insufficient context for decision-making. Long-horizon instructions such as "pass through the bedroom, locate the study, and pick up the book on the desk" require agents to chain subgoals while preserving spatial dependencies across rooms and corridors. Under partial observability, an agent's onboard camera captures only a limited field of view at each timestep, making historical context essential for inferring occluded landmarks, retracing steps, and maintaining spatial awareness. While Transformer-based Vision-Language Models (VLMs) (Vaswani et al., 2017b; Kim et al., 2021; Liu et al., 2023a; Li et al., 2023; Alayrac et al., 2022; Bai et al., 2023; Chen et al., 2024b) have achieved remarkable success in single-frame visual reasoning tasks such as Image Captioning (Hossain et al., 2019) and Visual Question

Answering (Antol et al., 2015), their adaptation to sequential tasks reveals fundamental limitations. Recent approaches in Vision-and-Language Action (Ma et al., 2024) and Navigation (Wu et al., 2024) tasks, including OpenVLA (Kim et al., 2024), RT-1 (Brohan et al., 2022), RT-2 (Brohan et al., 2023), Navid (Zhang et al., 2024), UniNavid (Zhang et al., 2025a), and NavGPT2 (Zhou et al., 2024a), have demonstrated the potential of VLMs in embodied scenarios. However, these methods struggle to efficiently integrate the extensive historical visual context necessary for long-horizon reasoning.

Existing strategies for integrating historical context into VLM backbones can be grouped into three main categories. (1) Token concatenation approaches are widely used in integrating historical context in Transformer-based models (Zhang et al., 2023a; 2024; Chen et al., 2021; Guhur et al., 2021; Lin et al., 2023). (2) Recurrent compression methods employ RNNs or LSTMs to compress the entire frame history into a single state vector (Krantz et al., 2020; Hong et al., 2020). (3) Previous studies also evaluate methods that maintain an external mapping and memory frameworks by external topological or semantic maps (Zhang et al., 2025b; An et al., 2023a;b; Chen et al., 2022a). Although these methods demonstrate effective performance, all three classes face certain limitations when applied to large-scale pretrained VLMs in environments that require context. Token concatenation disrupts the downstream token order and floods the model with redundant information. Recurrent compression lacks the capacity to represent fine temporal structure, leading to information loss over extended sequences. External memory methods depend on manually constructed maps that may not generalize across different environments. These limitations collectively highlight the necessity for a more efficient and effective methodology to integrate long historical context into pretrained VLMs.

Motivated by these challenges, in this study, we focus on eliminating memory bottlenecks and reducing computational complexity while preserving the original model architecture to maintain effectiveness. We draw insight from recent advances in parameter-efficient fine-tuning (PEFT) for large language models (LLMs) (Hu et al., 2022; Zhang et al., 2023b; Kim et al., 2025), which insert small trainable modules into frozen backbones with minimal overhead. This motivates our investigation of whether a similarly lightweight adapter paradigm can fuse rich historical visual information into VLMs while preserving their efficiency and pretrained knowledge. To this end, we propose the **D**ynamic **C**ontext **A**dapter (DCA), which compresses arbitrary sequences of past frame embeddings into a fixed set of learnable context vectors. DCA eliminates the memory bottleneck associated with naive token concatenation while capturing rich temporal semantics. To enable the model to consult its memory at every depth without altering original parameters or structure, these compressed representations are adapted to the LLM outputs through lightweight adapter modules and injected into each layer of the pretrained VLM. Our method delivers three key advantages. **First**, DCA ensures computational efficiency by maintaining constant input token length regardless of past frame quantity, achieving linear complexity growth with extended context. **Second**, DCA preserves fine-grained contextual details through dynamic compression of critical information into fixed context vectors. This approach avoids temporal detail loss common in recurrent models while removing redundant features. **Third**, DCA retains the original input and fully preserves the priors of the pretrained VLM, which enables maintaining its learned knowledge to the greatest extent possible.

To validate our approach, we analyze how DCA addresses the challenge of preserving rich temporal context in VLMs without architectural disruption and how it overcomes memory bottlenecks while maintaining fine-grained historical information for effective long-horizon VLN tasks. We evaluate DCA on the standard navigation benchmark and compare it against both RGB-only baselines and existing context-integration approaches. Our findings suggest that DCA resolves the core tension between capturing comprehensive visual history and maintaining computational tractability in partially observable environments. The experimental results demonstrate that DCA matches or exceeds prior methods in Success Rate while reducing attention FLOPs by over $25\%$ and cutting peak memory consumption by $15\%$ on long-horizon VLN tasks. Our contributions can be summarized as follows:

- We introduce DCA, an efficient and lightweight framework that addresses the core limitation of VLMs in sequential tasks by enabling dynamic compression and integration of historical visual context without disrupting pretrained model architecture or inflating input sequences.

- We demonstrate that DCA overcomes the fundamental challenge of information loss in recurrent-based approaches and memory explosion issues in concatenation methods, enabling VLMs to maintain rich temporal understanding across extended navigation episodes.

- We validate that DCA enables effective utilization of historical context for long-horizon reasoning in partially observable environments, achieving superior navigation performance.

## 2 RELATED WORKS

**Pretrained Vision-Language Models.** Large-scale VLMs (Kim et al., 2021; Alayrac et al., 2022; Liu et al., 2023a; Grattafiori et al., 2024; Touvron et al., 2023; Karamcheti et al., 2024; Abdin et al., 2024; Li et al., 2024; Zhang et al., 2023a; Bai et al., 2023; Chen et al., 2024b) have achieved impressive multimodal general-purpose reasoning capabilities. For example, ViLT (Kim et al., 2021) introduced a minimalist vision-language Transformer that forgoes region-based visual features for end-to-end image-text encoding. Likewise, LLaVA (Liu et al., 2023a) fine-tunes a pre-trained vision encoder together with a fine-tuned version of LLaMA (GenAI, 2023) using GPT-4 (Achiam et al., 2023) generated instruction follow-up data, producing a powerful multimodal assistant capable of open-ended visual dialogue. These models are typically designed for textual modality or static image-text pairs, and do not accommodate video or historical visual contexts essential for navigation and temporally extended reasoning tasks. LLaMA-VID (Li et al., 2024) extends the LLaMA (GenAI, 2023) for video-text tasks but primarily addresses the problem with naive inefficient token concatenation.

**Navigation with Pretrained Large Models.** Several recent works have explored applying foundation models to embodied VLN tasks. Intuitive approaches involve directly leveraging pretrained large language models as planners (Xu et al., 2023; Shah et al., 2023; Zhou et al., 2024b; Long et al., 2024; Chen et al., 2024a; 2025; Weerakoon et al., 2024), while other groups of works have shown great success in incorporating VLMs as navigation backbones (Zhou et al., 2024a; Lin et al., 2025; Pan et al., 2024; Liu et al., 2025; Zheng et al., 2024; Zhang et al., 2025a). NaVid (Zhang et al., 2024) fine-tunes a video-based VLM backbone to predict next-step actions by concatenating raw frame tokens, including both current and historical observations. The following work Uni-NaVid (Zhang et al., 2025a) unifies multiple navigation tasks in a video-based VLM backbone, processing long video streams end-to-end. Zhou et al. (2024a) augments LLMs with policy networks for VLN by input concatenation. While leveraging powerful pre-trained representations, they suffer from quadratic scaling with frame concatenation and lack mechanisms to distill and recall prior observations. DCA uses lightweight adapters to effectively decouple history context from LLM input, maintaining constant-length inputs while efficiently retrieving relevant historical information across layers.

**Historical Context in Navigation.** Traditional recurrent models maintained implicit memory via LSTM or GRU hidden states that carry over past perceptions (Anderson et al., 2018; Tan et al., 2019; Fried et al., 2018; Krantz et al., 2020; Song et al., 2024b; Hong et al., 2020; Krantz & Lee, 2022; He et al., 2024), but more recent approaches use Transformer-based architectures to capture longer-range dependencies (Majumdar et al., 2020; Zhang et al., 2025a; Song et al., 2024a; Lin et al., 2023; Guhur et al., 2021; Chen et al., 2021; Zhang et al., 2024). These methods integrate historical context by either maintaining recurrent hidden states or concatenating history frames as additional input during prediction, which may result in information loss. Other works have proposed building structured memory representations of the environment (Liu et al., 2023b; An et al., 2023a;b; Deng et al., 2020; Wang et al., 2023; 2024; Savinov et al., 2018; Chen et al., 2022b;a; Zhang et al., 2025b). Our work uses Transformer-based pretrained VLMs as backbones, but instead of adding additional input tokens, we introduce an efficient method to adapt context into LLM layers. Previous works were designed for static vision-language alignment tasks such as few-shot prompting or pretraining with fixed image-text pairs (Mañas et al., 2022; Radford et al., 2021). In contrast to these prior methods that perform one-time modality bridging, our model operates within a Partially Observable Markov Decision Process (POMDP) and must continuously compress an expanding observation history.

## 3 COMPUTATIONAL BARRIERS TO HISTORICAL CONTEXT INTEGRATION

The fundamental challenge in historical context integration lies in balancing temporal richness with computational feasibility. Examining the limitations of existing approaches reveals the computational barriers that prevent effective long-horizon reasoning in VLMs. Table 1 presents our analysis of computational costs across different integration strategies. Current methods face fundamental scalability issues that render them impractical for extended temporal contexts. **Concat-No-Adapt** concatenates all historical visual tokens and produces prohibitive LLM self-attention

Table 1: Complexity comparison (dominant terms). $T$=history frames, $p$=pooled tokens/frame, $S$=text length, $C/q$=memory/mapped tokens.

| Method | FLOPs (Visual Integration) | Self-Attn Context |
|---|---|---|
| Concat-No-Adapt | $\tilde{O}((S + Tp)^2 d)$ | $S + Tp$ (first layer) |
| Mapping-only (Mañas et al., 2022) | $\tilde{O}(qTpd)$ | $S + q$ (first layer) |
| DCA (ours) | $\tilde{O}(CTpd) + \tilde{O}(kSCd)$ | $S$ (all layers), cross-attn to $C$ |

cost of $\tilde{O}\big((S + Tp)^2 d\big)$ that scales quadratically with history length. **Mapping-only (Mañas et al., 2022)** maps $Tp$ tokens into $q$ tokens at cost $\tilde{O}(q \cdot Tp \cdot d)$, followed by first-layer self-attention over $(S + q)$ tokens. However, this approach inflates input context and restricts integration to a single layer. In contrast, our DCA addresses these limitations through efficient architectural design. The cross-attention mechanism in our compression module (detailed in Eq. (1) in Section 4.2.1) requires only $\tilde{O}(C \cdot Tp \cdot d)$ FLOPs and $O(Tp + C)$ memory tokens and achieves linear scaling in history length. Our multi-layer integration adds $\tilde{O}(k \cdot S \cdot C \cdot d)$ FLOPs while maintaining constant visual sequence length for self-attentions, avoiding quadratic explosion that plagues concatenation methods.

The following analysis reveals that existing methods face fundamental limitations in handling extended historical contexts: concatenation methods encounter quadratic computational explosion, while compression methods sacrifice temporal detail or impose architectural constraints. These computational barriers provide the foundation for our DCA, which we detail in the following section.

**Proposition 3.1 (Asymptotic scaling in $T$)** *Given fixed parameters $(C, S, k, d)$ and increasing history length $T$, our approach scales as $\tilde{O}(C \cdot Tp \cdot d)$ (compression) $+ \tilde{O}(k \cdot S \cdot C \cdot d)$ (injection) and exhibits linear complexity in $T$. Concat-No-Adapt scales quadratically in $(S + Tp)$, while mapping-only achieves linear compression $\tilde{O}(q \cdot Tp \cdot d)$ but inflates first-layer context to $(S + q)$.*

**Corollary 3.2 (Practical implications)** *When history length is substantial $(T \gg 1)$ and compression ratio satisfies $C \ll Tp$, our method delivers superior computational efficiency relative to concatenation approaches in both FLOPs and memory utilization. Compared to mapping-only methods operating under equivalent token budgets $(q = C)$, our approach maintains constant first-layer context while enabling stable multi-layer conditioning through gated cross-attention mechanisms.*

## 4 Efficient History Context Adaptation Methodology

### 4.1 Problem Formulation

We formulate efficient historical context integration as a computational optimization problem within the framework of VLN. The primary challenge lies in enabling VLMs to process extended temporal sequences while maintaining computational tractability as well as preserving pretrained knowledge.

**POMDP Formulation with Efficiency Constraints.** We model the task as a Partially Observable Markov Decision Process (POMDP) where computational efficiency becomes a primary constraint. At timestep $t$, the agent receives a natural language instruction $L_t$ and a visual observation sequence $X = \{X_1, X_2, \ldots, X_t\}$, where $X_{1:t-1}$ denotes historical frames and $x_t$ the current frame. Based on these inputs, the agent selects a low-level action $a_t \in A$ that transitions it into a new state with observation $X_{t+1}$. The observation space comprises monocular RGB images, and the action space includes qualitative action types and quantitative action arguments as established in VLN-CE (Krantz et al., 2020). VLN-CE environments feature complex visual occlusions and challenging long-horizon navigation goals where agents must recall and integrate multiple past observational frames for each critical decision. The efficiency challenge emerges from the requirement to process sequences of length $t$ that can extend to hundreds of frames in long-horizon scenarios. Conventional approaches face computational explosion as sequence length grows, making efficient context integration essential for practical deployment. This computational bottleneck motivates our investigation into developing a scalable context adapter that preserves rich historical information while maintaining efficiency.

**Efficient VLM Architecture Selection.** To maximize efficiency while demonstrating effectiveness, we employ a compact pretrained VLM backbone. Following recent advances in efficient VLM deployment (Zhang et al., 2024; 2025a;b; Kim et al., 2024), we adopt PrismaticVLM (Karamcheti et al., 2024) as our foundation. We select the phi-2+3b variant with only 3B parameters, which incorporates a ViT-based CLIP (Dosovitskiy et al., 2020; Radford et al., 2021) visual encoder, a lightweight Phi-2 (Abdin et al., 2024) language model, and multi-layer cross-modal projection. This architecture choice demonstrates that our efficiency gains extend beyond large-scale VLM models.

**Efficient Context Processing Pipeline.** Given visual observations $X$, we encode each frame into visual tokens and project them into a shared embedding space with language tokens. This process yields $X' = \{X'_{1:t-1}, X'_t\}$. Instructions $L_t$ are tokenized to produce $L'_t$. For action prediction at timestep $t$, we process current frame tokens $X'_t$ and instruction tokens $L'_t$ through the LLM while utilizing encoded historical frames $X'_{1:t-1}$ as inputs for efficient contextual embedding adaptation in LLM layers. This formulation establishes the computational constraints that our DCA must satisfy.

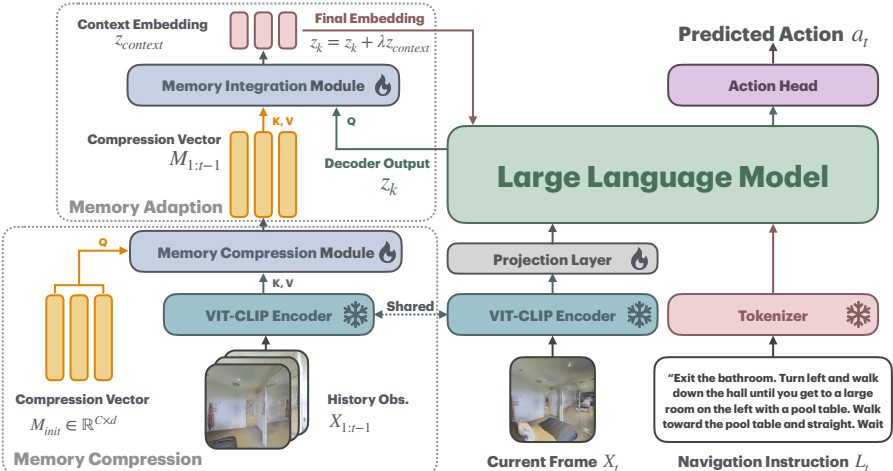

Figure 2: Model Architecture Overview. In each timestep t, the model receives inputs, including initial compression vector, history observations, current observations, and navigation instruction. We compress the historical context through a Memory Compressing Module, pass it to the Memory Integration Module, and adapt the resulting memory into the layer outputs of the LLM backbone.

### 4.2 DYNAMIC CONTEXT ADAPTATION (DCA) FOR EFFICIENT HISTORICAL INTEGRATION

Our DCA addresses the computational bottleneck of historical context integration through a two-stage architectural design that maintains linear complexity while preserving temporal richness. The core innovation lies in decoupling historical context processing from the main VLM backbone, which enables efficient memory management without sacrificing representational capacity. Given that navigation environments constitute POMDPs, agents must integrate previous observations for informed decision-making. However, naive concatenation of past tokens causes input sequence explosion and incurs super-linear self-attention costs (detailed in Section 4.3). DCA resolves this efficiency-accuracy trade-off by dynamically compressing historical context into a compact set of learnable memory vectors that adapt into LLM layers while preserving upstream pretrained semantics.

Fig. 2 illustrates our efficient dual-pipeline architecture. The standard VLM pathway processes $X_t$ through a shared visual encoder and tokenizes instruction $L_t$ via the Phi tokenizer. Both inputs pass through the pretrained VLM to produce decoder embedding $\mathbf{z}_t$, which an action head decodes into next-step action $a_t$ following standard next-token prediction. The efficiency-focused context adaptation pathway operates in parallel: a fixed-size learnable compression vector $\mathbf{M}_{\text{init}}$ queries past embeddings $X_{1:t-1}$ through our Memory Compression Module, producing compressed memory $\mathbf{M}_{1:t-1}$. Our Memory Integration Module then attends over $\mathbf{M}_{1:t-1}$ with current decoder queries to extract context-enhanced outputs that can adapt into LLM layers without inflating input sequences.

#### 4.2.1 EFFICIENT DYNAMIC CONTEXT VECTOR COMPRESSING

Our compression strategy achieves computational efficiency by transforming variable-length historical sequences into fixed-size representations while preserving critical temporal information. This design eliminates quadratic scaling in concatenation methods and enables practical deployment in resource-constrained scenarios. We initialize a learnable compression vector $\mathbf{M}_{\text{init}} = \texttt{nn.Embedding}(C, d).\texttt{weight} \in \mathbb{R}^{C \times d}$ for each timestep $t$, where $C$ denotes memory token count and $d$ represents embedding dimension. Historical frames $X_{1:t-1}$ are encoded via the vision encoder (Dosovitskiy et al., 2020; Radford et al., 2021) and concatenated to form encoded features $\mathbf{F}_{1:t-1} = \|_{t=1}^{t-1} \text{ViT-CLIP}(\mathbf{X}_i) \in \mathbb{R}^{(t-1) \times P \times d}$, where $P$ denotes image patch count. To reduce spatial redundancy, we apply grid pooling operator $\mathcal{G} : \mathbb{R}^{P \times d} \to \mathbb{R}^{p \times d}$ (with $p \ll P$) following established practices (Zhang et al., 2024; Li et al., 2024). This yields $\mathbf{F}_{1:t-1} = \mathcal{G}(\mathbf{F}_{1:t-1}) \in \mathbb{R}^{(t-1) \times p \times d}$. Our Memory Compression Module employs multi-layer cross-attention between $M_{\text{init}}$ and pooled features. We project $M_{\text{init}}$ as queries and history features as keys and values: $Q_M = M_{\text{init}} W_Q$, $K_F = \mathbf{F}_{1:t-1} W_K$, $V_F = \mathbf{F}_{1:t-1} W_V$. The compressed computation achieves $O(C \cdot p)$ complexity:

$$M_{1:t-1} = S_{\text{cps}} V_F \in \mathbb{R}^{C \times d}, \quad \text{where} \quad S_{\text{cps}} = \text{Softmax}(Q_M K_F^T) \in \mathbb{R}^{C \times p} \tag{1}$$

### 4.2.2 EFFICIENT CONTEXT ADAPTATION FOR LLM INTEGRATION

Our context adaptation strategy achieves computational efficiency by integrating compressed historical information directly into LLM layers without inflating input sequences or disrupting the original architecture. This approach maintains constant computational overhead regardless of history length while enabling multi-layer conditioning that enhances temporal understanding. The integration process operates on standard encoder-only multi-layer language models. For each layer $k$ with input $z_{k-1} \in \mathbb{R}^{S \times d}$, where $S$ represents sequence length, the standard layer output $z_k$ is formulated as:

$$z_k = \text{Atten}(Q_{k-1}, K_{k-1}, V_{k-1}), \quad Q_{k-1}, K_{k-1}, V_{k-1} = z_{k-1}W_Q^{k-1}, z_{k-1}W_K^{k-1}, z_{k-1}W_V^{k-1}. \tag{2}$$

Our Memory Integration Module enables efficient historical context adaptation into each Transformer layer. This module integrates the compressed context vector from Eq. (1) through lightweight cross-attention that maintains linear complexity. The module projects compressed historical context into key-value representations: $K_M = M_{1:t-1}W_K^M$ and $V_M = M_{1:t-1}W_V^M$. The context-enhanced output computation achieves efficiency by attending the original layer output $z_k$ to compressed historical vectors rather than processing full sequence history, which can be expressed as follows:

$$z_k^{\text{context}} = S_{\text{intg}}V_M, \quad \text{where} \quad S_{\text{intg}} = \text{Softmax}(Q_{k-1}K_M^T), \tag{3}$$

where $S_{\text{intg}}$ denotes the attention score of the integration module. The final layer output combines the context-enhanced representation with the original output through learnable scalar weighting as:

$$z_{k+1} \leftarrow z_{k+1} + \lambda z_{k+1}^{\text{context}}. \tag{4}$$

This design maintains computational efficiency by processing only $C$ compressed memory tokens per layer rather than the full history sequence of length $t$, achieving favorable $O(S \cdot C)$ for efficient context integration compared to the prohibitive $O(S \cdot t \cdot p)$ for the naive concatenation approaches.

### 4.3 ON THE LINEAR SCALABILITY TO EXTENDED HISTORICAL CONTEXT

The efficiency of historical context integration becomes critical when processing extended temporal sequences. We analyze the complexity characteristics of our approach compared to standard concatenation methods to demonstrate the efficiency advantages that enable practical deployment. Consider the attention sublayer of a typical Transformer decoder with textual context length $N_{\text{text}}$ and visual token length $tN_v$ at timestep $t$, where $N_v$ represents tokens per frame and $t-1$ denotes history frame count. Standard LLMs process total context length $N_{\text{text}} + tN_v$ directly, where $(t-1)N_v \gg N_{\text{text}}$ in long-horizon scenarios. This approach incurs quadratic complexity $\mathcal{O}((N_{\text{text}} + tN_v)^2)$ for self-attention operations, creating severe memory and computational overhead as history length increases.

Our compression mechanism transforms this computational bottleneck through efficient architectural design. The proposed Memory Compression Module (Fig. 2) compresses history frames to fixed length $C$ with complexity $\mathcal{O}((t-1)N_vC)$. This enables the LLM to process only $N_{\text{text}} + N_v$ tokens directly while integrating compressed information via cross-attention at complexity $\mathcal{O}((N_{\text{text}} + N_v)C)$ per layer. Together, for an $L$-layer model, the total inference cost can be expressed as follows:

$$\mathcal{O}(\ \underbrace{L(N_{\text{text}} + N_v)^2}_{\text{LLM context self-attention}} + \underbrace{L(N_{\text{text}} + N_v)C}_{\text{Memory integration (each layer)}} + \underbrace{(t-1)N_vC}_{\text{Memory compression}} \ ) \tag{5}$$

This design achieves linear scaling with history length $(t-1)N_v$ compared to the quadratic baseline complexity $\mathcal{O}(L(N_{\text{text}} + tN_v)^2)$ that grows substantially through concatenation. The linear scaling characteristic enables efficient processing of long-horizon tasks with extensive visual history while maintaining computational tractability and consistent performance across varying sequence lengths.

## 5 EXPERIMENTAL RESULTS

In this section, we validate the efficiency and effectiveness of the proposed context adapter through organized experiments on three topics: (1) Efficiency: compute overhead (FLOPs), training resource requirements, and inference latency compared against approaches that incorporate history by concatenating past frames as additional tokens; (2) Effectiveness: how well the method utilizes pretrained VLM semantics while maintaining navigation performance across various history lengths, and (3) Design Insights: the specific design choices that contribute most to the gains and how they interact.

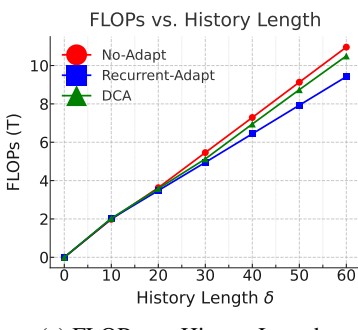 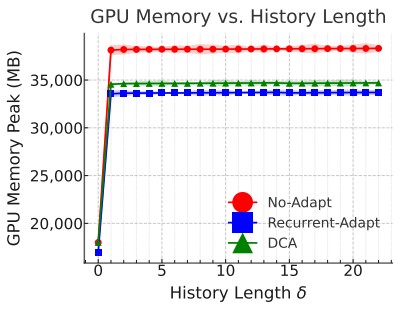

(a) FLOPs vs. History Length        (b) GPU Memory vs. History Length

Figure 3: Computational efficiency analysis of context-adaptation methods. **Left:** FLOP requirements as history length increases. **Right:** Peak GPU memory consumption across varying history lengths.

## 5.1 EXPERIMENTAL SETUP

**Baselines.** For a fair comparison, we evaluate methods that implement end-to-end learning with low-level action primitives in the VLN-CE environments. **(1) Seq2Seq (Krantz et al., 2020)**: A recurrent sequence-to-sequence architecture that directly maps RGBD observations to navigation actions. The RGB-Seq2Seq variant processes RGB inputs exclusively. **(2) CMA (Krantz et al., 2020)**: Implements cross-modal attention between instructions and RGBD observations for action prediction. Note that RGB-CMA denotes the RGB-only configuration. **(3) NaVid (Zhang et al., 2024)**: Employs a frozen VLM backbone to formulate navigation as next-token prediction over RGB sequences. This method concatenates historical observations as additional language tokens and applies auxiliary training objectives. NaVid-IL represents the imitation learning configuration. For efficiency experiments, we establish two controlled baselines that share our VLM backbone and training protocol: **(1) No-Adapt**: Processes historical frames as additional VLM input tokens without compression or adaptation mechanisms. **(2) Recurrent-Adapt**: Replaces our Memory Compression Module with an LSTM that sequentially processes past frame embeddings into fixed-size context representations while maintaining the identical backbone architecture as well as the training pipeline.

**Simulation Environment.** The models are trained on R2R (Anderson et al., 2018) dataset under continuous setting as in VLN-CE (Krantz et al., 2020), where the agent is required to navigate in unseen continuous environments by predicting discrete actions, VLN-CE contains 146304 episodes across 60 scenes, adapted from Tan et al. (2019). Similar to prior settings (Anderson et al., 2018; Krantz et al., 2020), we leverage several representative metrics for evaluating the navigation performace: success rate (SR), success rate weighted by the ratio between the shortest path length and the predicted path length (SPL), oracle success rate (OSR), trajectory length (TL), as well as navigation error (NE).

**Implementation Details.** Our method utilizes pre-trained VLM from PrismaticVLM (Karamcheti et al., 2024). Following Kim et al. (2024); Zhang et al. (2024; 2025a), we froze the vision encoder and finetune the LLM and projection layers. Models are trained by FSDP sharding strategy under Imitation Learning (IL) by the oracle trajectories across 8 NVIDIA L40 with 48GB memory each. The loss is computed by a standard SoftMax IL loss. Memory Compressing Module and Memory Integration Module both implement multi-head multi-layer attention introduced in work proposed by Vaswani et al. (2017a). We set $\lambda = 1$ mentioned in Eq. (4) and $C = 128$ defined in Section. 4.2.1.

## 5.2 ANALYSIS ON MODEL EFFICIENCIES

**Computational Efficiency Analysis.** Table 2 presents a comparison of inference throughput across methods. Our DCA approach demonstrates substantial efficiency gains compared to the No-Adapt baseline: average inference time decreases from 3.21s to 2.71s per step, FLOPs reduce from 4.77T to 4.23T, and peak GPU memory usage drops from 37.84 GB to 34.31 GB. These improvements directly result from our efficient dual-pipeline architecture that decouples historical context processing from the main VLM backbone. To analyze scalability characteristics, Fig. 3 illustrates computational overhead as history length $\delta$ increases across methods. While all approaches exhibit approximately linear growth due to feed-forward network dominance independent of $\delta$, critical efficiency distinctions emerge with extended sequences. At initialization ($\delta = 0$), all methods demonstrate comparable FLOP requirements. However, as history length increases, the No-Adapt baseline (red curve) exhibits the steepest computational growth. At $\delta = 30$, DCA achieves over 25% reduction in additional FLOPs relative to No-Adapt, validating our architectural efficiency claims. The Recurrent-Adapt baseline presents an interesting contrast: it demonstrates the most favorable FLOP scaling due to

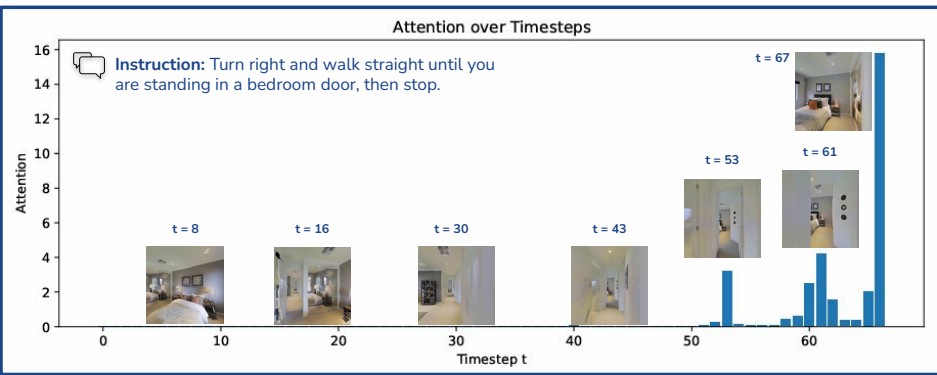

Figure 4: Averaged attentions of the Memory Compression Module across navigation timesteps for unseen evaluation episode 37 in VLN-CE, with the corresponding visual observations indicated.

minimal per-timestep recurrent update costs. However, this apparent efficiency advantage comes at the expense of representational capacity, ultimately limiting navigation performance. This trade-off illustrates the fundamental challenge our method addresses: achieving both computational efficiency and representational richness for effective historical context integration. Our results demonstrate that DCA can effectively resolve this efficiency-accuracy tension through efficient architectural design rather than sacrificing either computational tractability or its temporal understanding capabilities.

**Memory Efficiency.** Fig. 3 (b) presents GPU memory consumption patterns during training as history length $\delta$ varies. All methods exhibit modest memory growth with increasing $\delta$, as model weights dominate total memory usage. At baseline ($\delta = 0$), Recurrent-Adapt shows the lowest memory requirements, while No-Adapt and DCA present nearly identical consumption.

Table 2: Inference Throughput Comparisons.

| Method (Input: RGB) | # Params | Step Inf. Time | FLOPs (T) | Mem. Peak (GB) |
|---|---|---|---|---|
| Navid-IL | 7B | 2.86 | 4.89 | 48.61 |
| No-Adapt | **3B** | 3.21 | 4.77 | 37.84 |
| Recurrent-Adapt | **3B** | **2.50** | **4.14** | 35.65 |
| DCA (Ours) | **3B** | 2.71 | 4.23 | **34.31** |

This similarity confirms that our DCA module introduces minimal architectural overhead. However, a significant efficiency gap emerges for $\delta \geq 1$: DCA consistently uses approximately 30% less memory than No-Adapt. This reduction directly results from our compressed context representation strategy, which processes fixed-size memory vectors rather than expanding token sequences. The memory efficiency advantage becomes increasingly pronounced with longer histories, demonstrating the practical benefits of our compression-based approach for resource-constrained deployment scenarios.

## 5.3 EVALUATIONS ON VLN PERFORMANCE

Table 3 presents navigation performance on VLN-CE R2R Val-Unseen split. Methods are organized by input modality: approaches using additional sensors beyond RGB (#1-#8) appear above the first horizontal line, while RGB-only methods (#9-#15) are grouped below. NaVid variants receive separate categorization due to auxiliary co-training protocols. DCA shows substantial performance improvements under the low-level action VLN-CE framework. Compared to recurrent baselines RGB-Seq2Seq and RGB-CMA, DCA achieves relative success rate improvements of 13.7% and 8.7%, respectively. Against

Table 3: Evaluations on VLN-CE R2R Val-Unseen. *: Methods use high-level action space. †: Methods apply the waypoint predictor proposed in Hong et al. (2022). ‡: Methods use extra visual data than MP3D scenes Chang et al. (2017).

| # | Method | Observation | | | | VLN-CE R2R Val-Unseen | | | | |
|---|---|---|---|---|---|---|---|---|---|---|
| | | Pan. | S.RGB | Depth | Odo. | TL | NE↓ | OS↑ | SR↑ | SPL↑ |
| 1 | HPN+DN* Krantz et al. (2021) | ✓ | | ✓ | ✓ | 7.62 | 6.31 | 40.0 | 36.0 | 34.0 |
| 2 | CMA*† Hong et al. (2022) | ✓ | | ✓ | ✓ | 10.90 | 6.20 | 52.0 | 41.0 | 36.0 |
| 3 | RecurrentVLN*† Hong et al. (2022) | ✓ | | ✓ | ✓ | 12.23 | 5.74 | 53.0 | 44.0 | 39.0 |
| 4 | Sim2Sim* Krantz & Lee (2022) | ✓ | | ✓ | ✓ | 10.69 | 6.07 | 52.0 | 43.0 | 36.0 |
| 5 | HAMT*†‡ Chen et al. (2021) | ✓ | | ✓ | ✓ | – | 4.80 | – | 55.0 | 51.0 |
| 6 | LAW Raychaudhuri et al. (2021) | | ✓ | ✓ | ✓ | 8.89 | 6.83 | 44.0 | 35.0 | 31.0 |
| 7 | Seq2Seq Krantz et al. (2020) | | ✓ | ✓ | | 9.30 | 7.77 | 37.0 | 25.0 | 22.0 |
| 8 | CMA Krantz et al. (2020) | | ✓ | ✓ | | 8.64 | 7.37 | 40.0 | 32.0 | 30.0 |
| 9 | NaVid Zhang et al. (2024) | | ✓ | | | 7.63 | 5.47 | 49.1 | 37.4 | 35.9 |
| 10 | NaVid-IL Zhang et al. (2024) | | ✓ | | | – | 7.10 | 20.6 | 14.4 | 12.4 |
| 11 | RGB-Seq2Seq Krantz et al. (2020) | | ✓ | | | 4.86 | 10.10 | 8.10 | 0.00 | 0.00 |
| 12 | RGB-CMA Krantz et al. (2020) | | ✓ | | | 6.28 | 9.55 | 10.80 | 5.00 | 4.43 |
| 13 | DCA (No-Adapt) | | ✓ | | | 3.91 | 7.12 | 8.86 | 7.23 | 7.00 |
| 14 | DCA (Recurrent-Adapt) | | ✓ | | | 8.44 | 9.56 | 7.14 | 6.59 | 5.44 |
| 15 | **DCA** | | ✓ | | | 6.73 | **6.77** | **25.3** | **13.7** | **12.9** |

Recurrent-Adapt, which shares our backbone and adaptation framework, DCA delivers 7.11% SR improvement, validating dynamic compression effectiveness over recurrent approaches. DCA outperforms concatenation-based approaches: it surpasses No-Adapt by 6.47% in SR while matching NaVid-IL performance despite using a smaller backbone (3B vs. 7B parameters) and standard training rather than auxiliary co-training. The competitive Oracle Success (OS) Rate demonstrates effective instruction comprehension. These results establish DCA's superior efficiency-performance trade-offs.

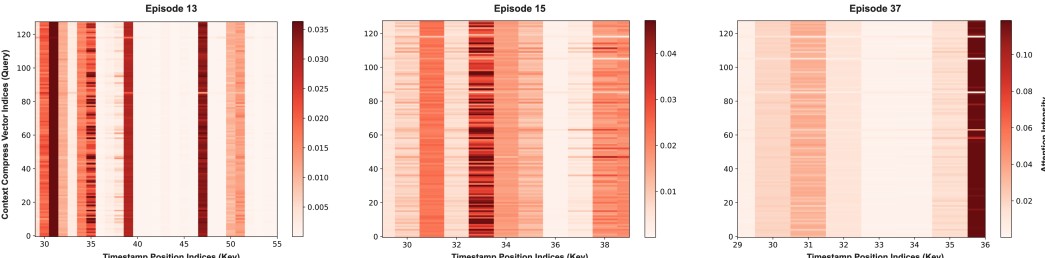

Figure 5: Attention heat map of the Memory Compression Module with initial compression vectors as queries and encoded historical frames as keys, indexed by timesteps. Please note that the initial 30 timesteps are truncated for improved visualization clarity across three representative episodes.

### 5.4 VISUALIZATIONS OF DYNAMIC CONTEXT COMPRESSION

To understand our compression efficiency mechanisms, we analyze attention patterns within the Memory Compression Module to identify which historical frames contribute most significantly to navigation decisions. This reveals how our method achieves computational efficiency through selective temporal prioritization rather than uniform compression. Fig. 4 presents attention visualization for our compression module. Following Eq. (1), we compute per-head attention scores $S_{\text{cps}}$ across the trajectory at final navigation step $T = 68$. These scores weight frame features from $t = 0$ to $t = 67$ during compression. Aggregating scores across heads produces a per-timestep attention profile demonstrating temporal selection priorities. The analysis reveals selective focus on semantically relevant observations, validating our compression approach. Early observations without visible targets (bedroom door) receive negligible attention weights, reflecting limited utility for decision-making at $T = 68$. Conversely, frames containing critical visual cues exhibit pronounced attention peaks: the target door at $t = 53$ and $t = 61$, and bedroom interior at $t = 67$ show substantially elevated weights. Fig. 5 confirms concentrated focus on later frames where goal locations become visible. These patterns confirm our method's ability to identify and prioritize critical contextual features while efficiently discarding temporally irrelevant information, achieving efficiency through intelligent temporal filtering rather than indiscriminate reduction. Additional analyses appear in Appendix B.

### 5.5 ABLATION STUDIES

Table 4 analyzes design choices of our method. We compare our default context injection via Eq. (3), which adds compressed context with learnable coefficient $\lambda$, against FiLM-based fusion applying $z_{k+1} \leftarrow z_{k+1} + (\alpha\, z_{k+1}^{\text{context}} + \beta)$ with zero-initialized parameters $\alpha$ and $\beta$ following Kim et al. (2025). FiLM fusion shows substantial SR and SPL drops because additional scaling parameters hinder stable context integration. Varying $\lambda$ (0.5, 0.8) shows larger values

Table 4: The ablative investigation on feature adaptation, context compression, and compression vector length.

| | | VLN-CE R2R Val-Unseen | | | | |
|---|---|---|---|---|---|---|
| | Type | TL | NE↓ | OS↑ | SR↑ | SPL↑ |
| Feature Fusing | FiLM Adapting | 2.32 | 11.4 | 6.25 | 5.47 | 5.26 |
| | $\lambda = 0.5$ | 7.34 | 7.21 | 14.6 | 10.12 | 9.66 |
| | $\lambda = 0.8$ | 6.59 | 7.01 | 17.8 | 11.4 | 9.54 |
| Context Compression | Instruction Attention | 7.23 | 6.90 | 8.86 | 7.23 | 6.99 |
| | $C = 24$ | 12.1 | 12.3 | 8.12 | 6.94 | 5.12 |
| | $C = 48$ | 10.5 | 11.4 | 10.27 | 6.88 | 5.64 |
| | $C = 64$ | 8.30 | 8.64 | 17.6 | 9.23 | 8.77 |
| **Ours** | Full Setting | 6.73 | **6.77** | **25.3** | **13.7** | **12.9** |

consistently improve success and SPL, confirming the importance of weighted historical context. For compression designs, we augment the compression module with cross-attention over instruction embeddings, hypothesizing context relevance correlates with instruction semantics. This variant underperforms direct compression due to data quality issues in R2R where instructions and trajectories are misaligned. Additionally, we examine the effect of memory capacity by varying memory tokens $C$ (24, 48, 64). Performance improvements correlate with increased $C$ values. This finding indicates that greater capacity captures richer temporal patterns, though excessive increases risk overfitting.

## 6 CONCLUSION

In this study, we introduced DCA, a lightweight framework that efficiently integrates historical context into pretrained VLMs without inflating input token lengths. Our proposed approach employed a Memory Compression Module to distill past frame embeddings into fixed-size learnable memory vectors and a Memory Integration Module to adapt these compressed representations into each Transformer layer. This design preserved the pretrained VLM architecture while achieving linear scaling with extended context lengths. Our extensive evaluations on downstream VLN tasks demonstrated that DCA can achieve superior efficiency-performance trade-offs compared to existing approaches.

## 7 REPRODUCIBILITY

We made our code publicly accessible in `https://anonymous.4open.science/r/diffuser_navigator-0670/README.md`.

## 8 ETHICS STATEMENT

This work adheres to the ICLR Code of Ethics. In this study, no human subjects or animal experimentation was involved. All datasets used, including {R2R Anderson et al. (2018)}, were sourced in compliance with relevant usage guidelines, ensuring no violation of privacy. We have taken care to avoid any biases or discriminatory outcomes in our research process. No personally identifiable information was used, and no experiments were conducted that could raise privacy or security concerns. We are committed to maintaining transparency and integrity throughout the research process.

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

# A    TRAINING DCA

## A.1    TRAINING PARAMETERS

We present the full list of the hyper-parameters used in our experiments in Table. 5.

Table 5: Hyper-parameters used in our experiments.

| Hyper-parameter | Value |
| --- | --- |
| Optimizer | AdamW |
| Base learning rate | $2.5 \times 10^{-5}$ |
| Learning-rate scheduler | CosSchedule + Warmup |
| Warmup steps | 10% of total steps |
| FSDP Sharding Strategy | FULL_SHARD |
| Epochs | 24 |
| Batch size (Global) | 32 |
| Total training steps | 360,000 |
| Dropout rate | 0.1 |
| Initial Compression Vector Length $C$ | 128 |
| Adaptation Coeficcient $\lambda$ | 1.0 |
| Number of Layers in Compression Module | 8 |
| Number of Layers in Adaptation Module | 8 |
| Number of Attention Heads in Compression Module | 4 |
| Number of Attention Heads in Adaptation Module | 4 |
| Hidden embedding size | 2560 |
| Gradient clipping norm | 1.0 |
| Mixed-precision training | bFP16 |
| Random seed | Same as VLN-CE Krantz et al. (2020) |

## A.2    DATASET & SAMPLING

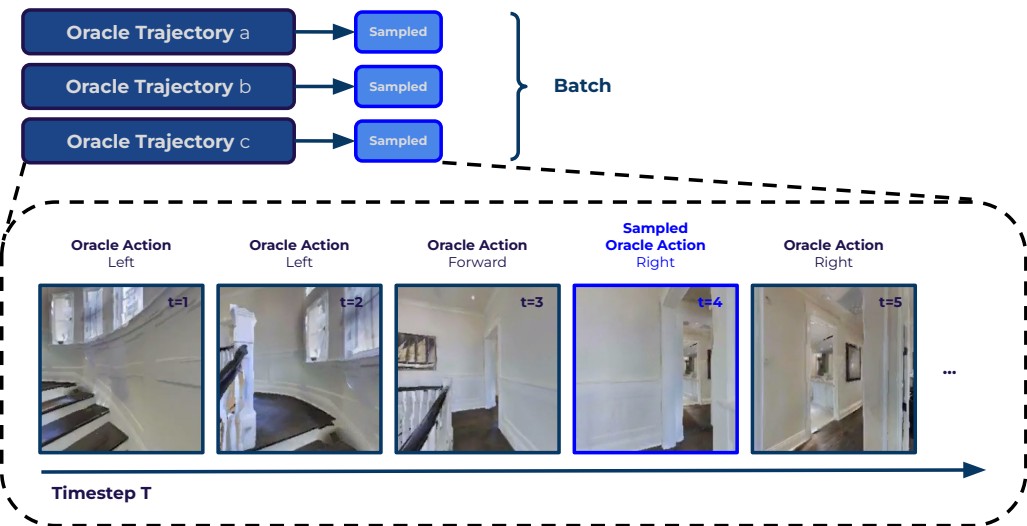

Figure 6: Illustration of Data Sampling of DCA Training.

During training, we collect $15,000$ oracle trajectories from VLN-CE Krantz et al. (2020) augmented training split Tan et al. (2019). Considering the fact that oracle navigation action distributions are not originally balanced in the oracle trajectory (e.g. there is only 1 <STOP> action in each individual oracle trajectory, but  50 other actions according to Krantz et al. (2020)), to ensure effective training and a better in-sample correlations. During training, given $n$ collected oracle trajectories with observations and ground-truth actions, we sample a single timestep from each of these trajectories, resulting in $n$ timesteps as a batch input to our model, we also record their history observations as

context input, this process is illustrated in Fig. 6. We sample action $a$ in each trajectory according to the following categorical probability distribution $\pi$, which reasonably scales the original data occurrence probability for each action:

$$\pi(a) = \begin{cases} 0.10, & a = \text{stop}, \\ 0.40, & a = \text{forward}, \\ 0.25, & a = \text{left}, \\ 0.25, & a = \text{right}, \end{cases} \qquad a \sim \text{Categorical}(\pi).$$

## B  ADDITIONAL QUALITATIVE RESULTS

Our qualitative analysis employs a series of representative navigation episodes shown in Fig. 7, Fig. 8, Fig. 9 and Fig. 10. In each case, the left panel presents a top-down map overlaid with the agent's executed trajectory, blue lines are the agent actual trajectory, where the dotted green lines are the expected trajectory. On the right panel displays a histogram of attention weights assigned by the compression module at the final decision step. Larger bars in the attention histogram correspond to higher attention scores for the associated past frames. This inspection reveals that the compression module consistently concentrates on frames containing task, relevant cues such as doorways, corridor intersections and target landmarks. Even when an episode spans over ten meters of travel, the method highlights only a handful of critical observations rather than processing every frame equally. Such selective focus confirms that our adapter can distill essential information from long visual histories without concatenating the entire sequence.

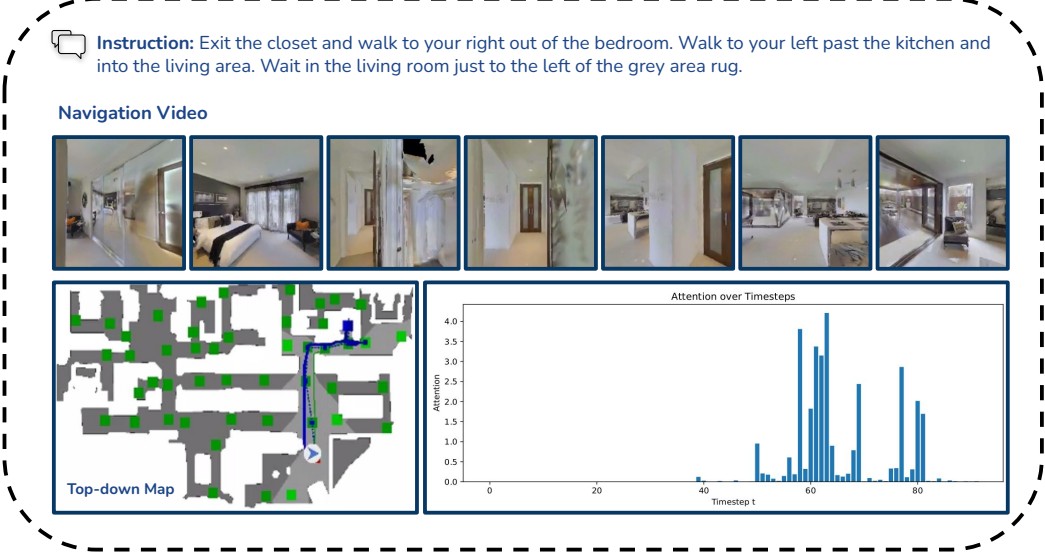

Figure 7: Trajectory Summery of Episode 38.

## C  LIMITATIONS & FUTURE WORKS

While the our method delivers notable efficiency gains alongside competitive navigation results, demonstrating impressive performance-efficiency trade-offs. However, compressing visual histories of varying length into a fixed-size memory vector may obscure subtle temporal cues on long trajectories, which can undermine performance on tasks requiring precise step by step reasoning. Additionally, our experiments focus on a relatively small VLM backbone, so the adapter's behavior on more recent, large scale models remains an open question. Moreover, the current design assumes a Transformer architecture in which visual and linguistic features occupy a shared representation space; extending the method to architectures with fundamentally different structures or to other modalities would demand a substantial redesign of both adapter components and the compression workflow.

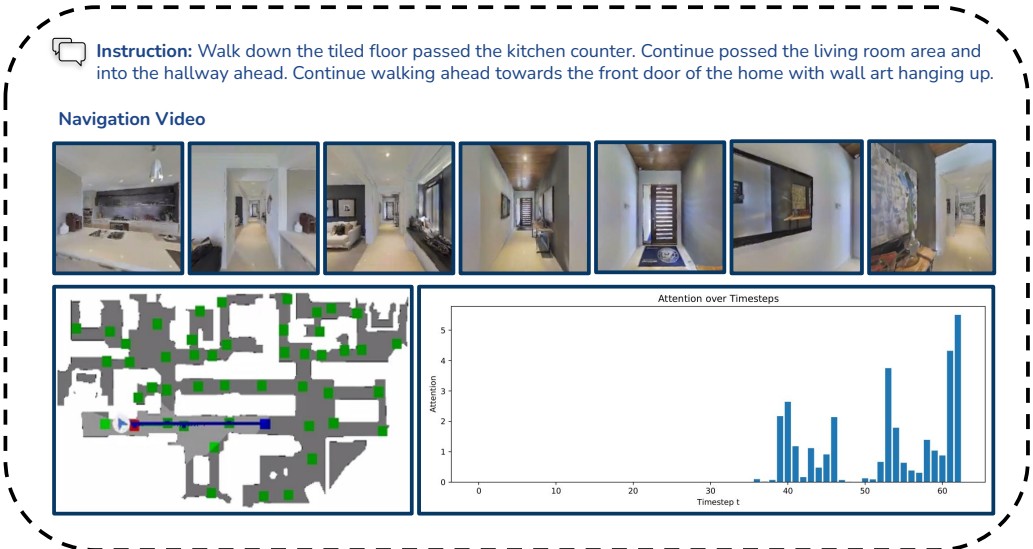

Figure 8: Trajectory Summery of Episode 137.

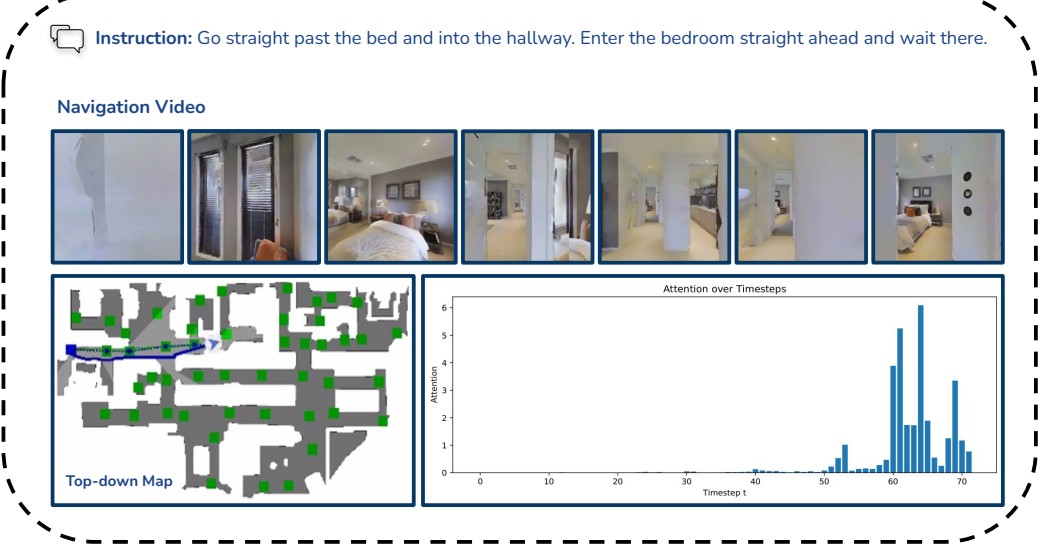

Figure 9: Trajectory Summery of Episode 368.

We hope our approach opens several avenues for future researches on leveraging pre-trained VLMs into downstream tasks. Integrating richer spatial priors into the compression stage, exploring adaptive memory capacities that vary with environment complexity, and extending DCA to other sequential multimodal tasks, such as video question answering or long-horizon robotic manipulation, represent promising directions to further harness historical context in large-scale pretrained models.

## D    RELATION TO FEATURE-MAPPING ADAPTERS

Mapping-based adapters—such as MAPL (Mañas et al., 2022) which learn a transformer mapping from visual tokens to LLM-consumable embeddings, which aggregate many visual tokens into a few query tokens, offer efficient frame-/short-window token reduction injected at the first LLM layer. In contrast, our memory compression performs query-guided, history-spanning cross-attention over all past frames, producing $C$ memory tokens $M_{1:t-1} \in \mathbb{R}^{C \times d}$ that are gated and integrated at multiple top layers. Formally, let pooled per-frame features be $F \in \mathbb{R}^{(t-1) \cdot p \times d}$, and learnable

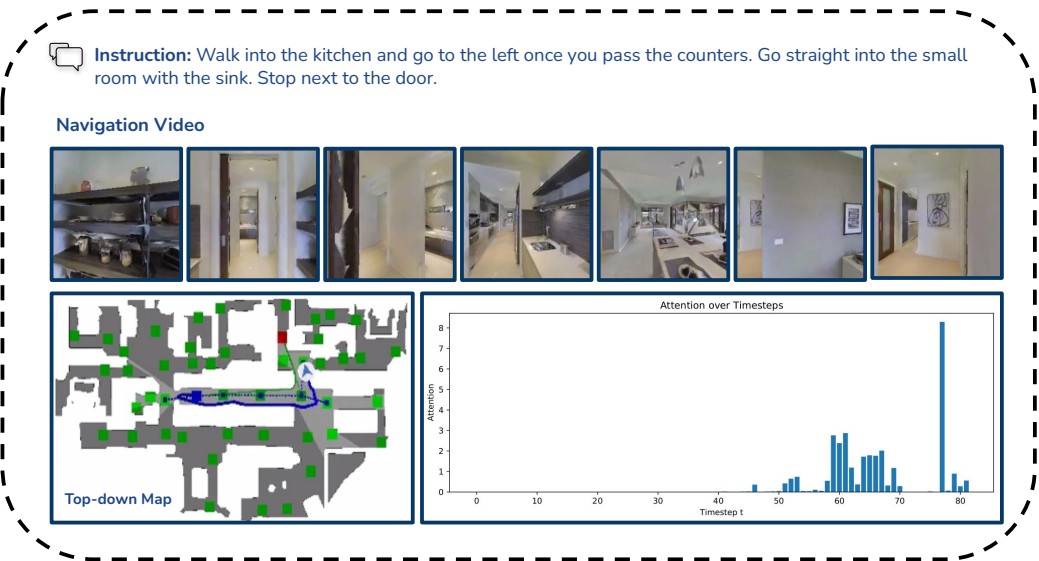

Figure 10: Trajectory Summery of Episode 361.

queries $Q_M \in \mathbb{R}^{C \times d}$; then

$$M_{1:t-1} \;=\; \text{softmax}\!\left(\frac{Q_M K_F^{\mathsf{T}}}{\sqrt{d}}\right) V_F, \tag{6}$$

where $(K_F, V_F) \in \mathbb{R}^{(t-1)\cdot p \times d}$ are key/value projections of $F$. The integration at LLM layers $\ell \in \mathcal{L}$ adopts a gated cross-attention:

$$H_\ell \leftarrow (1 - \lambda_\ell)\, H_\ell \;+\; \lambda_\ell\, \text{XAttn}(H_\ell,\, M_{1:t-1}), \qquad \lambda_\ell \in [0, 1], \tag{7}$$

with either fixed or learnable $\lambda_\ell$. We next state equivalence conditions clarifying when memory compression degenerates to mapping-only.

**Proposition D.1 (Degenerate equivalence to mapping-only)** *Assume: (i) no temporal adaptivity (the attention weights in equation 6 are independent of time and computed on a single frame or a fixed small window), (ii) single-layer injection ($\lambda_\ell = 0$ for all but the first LLM layer), and (iii) the number of memory tokens $C$ equals the number of mapped tokens q used by a mapping network. Then equation 6–equation 7 are functionally equivalent to a mapping-then-first-layer-injection scheme (MAPL/QPMapper-type) up to a reparameterization of the query projections. Conversely, if any of (i)–(iii) is violated—in particular, if attention is temporally adaptive over full history or if multi-layer gated integration is used—the equivalence does not hold.*

*Proof sketch* When (i)–(iii) hold, equation 6 reduces to a query-guided convex combination of within-window tokens, identical in form to attention-based token resampling. With equation 7 restricted to the first layer and constant $\lambda$, the overall computation matches mapping-only injection. Temporal adaptivity couples attention to history-dependent features; multi-layer gating composes multiple residual cross-attentions, which cannot, in general, be folded into a single first-layer injection. $\qquad\square$

