# OpenReview forum: "Dynamic Context Adapters: Efficiently Infusing History into Vision-and-Language Models"
_ICLR.cc/2026/Conference — ICLR 2026 Conference Withdrawn Submission_

### Official Review · Reviewer_VLzi · 2025-10-30

**Soundness:** 2
**Presentation:** 2
**Contribution:** 2
**Rating:** 2
**Confidence:** 5

**Summary:**

This paper addresses a critical challenge in applying Vision-and-Language Models (VLMs) to sequential decision-making tasks: the efficient integration of historical context. The authors argue that existing methods are flawed. The standard approach of concatenating past visual frames to the input of a Transformer-based VLM leads to quadratic growth in computational complexity and memory usage, making it infeasible for long-horizon tasks. Alternative methods, such as compressing history with recurrent models (LSTMs/RNNs), often suffer from significant information loss, while external memory-based approaches can be difficult to generalize.
To solve this, the authors propose the Dynamic Context Adapter (DCA), a lightweight and efficient module designed to work with pretrained VLMs. The core idea of DCA is to decouple the processing of historical context from the main VLM pipeline. It operates in two stages:
1. Memory Compression: A Memory Compression Module takes all past visual observations, encodes them, and uses a multi-layer cross-attention mechanism to compress this variable-length history into a fixed-size set of learnable "memory vectors." This process scales linearly with the length of the history, avoiding the quadratic explosion of concatenation.
2. Memory Integration: A Memory Integration Module then injects this compressed historical context into each layer of the pretrained Language Model (LLM) backbone. This is done via another lightweight cross-attention mechanism where the LLM's internal states attend to the compressed memory vectors.

This design allows the model to maintain a constant input token length for the main VLM, preserving its computational efficiency, while still giving it access to a rich, dynamically compressed summary of the past. The authors validate their approach on the challenging Vision-and-Language Navigation in Continuous Environments (VLN-CE) task. Their experiments show that DCA significantly reduces computational overhead (over 25% fewer attention FLOPs and 13% less memory) compared to naive concatenation, while simultaneously improving navigation performance over both concatenation-based and recurrent-based baselines.

**Strengths:**

1. The paper addresses an important problem of processing long visual sequences efficiently and the authors propose a reasonable approach for compressing prior context into a constant length.
2. The authors present thorough evaluations, ablations and analysis, including the computational requirements of the method and insightful visualizations of the Memory Compression Module's attention (Figures 4 and 5).

**Weaknesses:**

1. The authors have completely missed a long line of related literature that tries to decrease the computational requirements in long video processing for understanding and/or generation, use a memory module and/or memory compression techniques. It is impossible to understand the contributions of the method when it is not compared to any related literature and no comparisons are being drawn. As a few points, authors should check papers such as: BigBird and LongFormer in LMs, Transformer-XL, Retro, Compressive Transformer, ∞-former, MemDPC, MeMViT, LongMem, MC-ViT, Mirasol3B, and even more recent papers.
2. Also, including more video understanding benchmarks (such as EgoSchema, Next-QA) might make the comparisons more complete.

**Questions:**

1. How is this work compared to all related literature on long video processing/understanding (see references above)?

---

### Official Review · Reviewer_FNV5 · 2025-11-01

**Soundness:** 1
**Presentation:** 1
**Contribution:** 2
**Rating:** 2
**Confidence:** 4

**Summary:**

This paper solves the long context problem of VLMs (also called the long horizon in embodied and other agent problems). To solve the problem, the authors proposed an attention-based method, where they use a pre-defined set of learnable key-vectors to compress the long-context input of VLMs. The evaluation on the Vision-Language Navigation (VLN) task shows that the proposed method is proper for context length management.

**Strengths:**

The paper studies a very important problem of VLMs: how to process long-context input.

**Weaknesses:**

1. The motivation is not clear. The authors listed three kinds of previous works between lines 66 and 73 (the Introduction section). However, how those methods works are not well explained. For example, why does token concatenation have the problem of redundant information? There are a lot of works about long-context [1,2], and obviously, those long-context inputs don't have redundant information.

Seems that the authors assume all readers of their paper know the references of their paper perfectly, which makes their paper now self-included.

2. Besides the introduction, the paper has a lot of parts that are too dense, including lines 152 - 159 (Table 1), 263, and 277-278. For example, in the caption of Table 1, the authors only briefly introduce those variables (T, p, S, C/q). Even worse, a few variables introduced here are specific to their tasks, which are common in embodied tasks, like mapped tokens. Those short and brief definition further makes their paper less self-contained. The paper is hard to understand, even for embodied researchers.

3. The authors listed token concatenation as a non-working method for long-horizon tasks in VLN. However, there are a lot of works studying the long-context problem of VLM, which can perfectly solve the problem studied in this paper. However, the authors didn't discuss this line of work.

Meanwhile, the authors talk about how the token concatenation can flood current VLMs. However, they didn't discuss how many tokens are in a concatenated case. The GPT-5 and Gemini 2.5 pro support 400K and 2M tokens. Are your input longer than this?

4. Motivation and evaluation are mismatched: The motivation in the abstract of this paper is about processing the long input for VLMs. However, the evaluation is only about vision-language navigation (VLN), a task for the embodied agent. Meanwhile, numerous long-context tasks [1, 2] have been proposed recently.

[1] Wang, Zhaowei, et al. "MMLongBench: Benchmarking Long-Context Vision-Language Models Effectively and Thoroughly." NeurIPS 2025.
[2] Wang, Weiyun, et al. "Needle in a multimodal haystack." NeurIPS 2024.

**Questions:**

Seed the weakness

---

### Official Review · Reviewer_B64u · 2025-11-01

**Soundness:** 3
**Presentation:** 3
**Contribution:** 2
**Rating:** 4
**Confidence:** 3

**Summary:**

The paper introduces the Dynamic Context Adapter (DCA) for integrating historical temporal vision information into pre-trained VLMs for sequential decision-making tasks like navigation. The memory compression module compresses all past visual observations to a fixed-size set of compressed context vectors. The experimental results show that DCA outperforms concatenation- and recurrent-based methods.

**Strengths:**

The method has a better trade-off between efficiency and accuracy compared to concatenating and recurrent methods.

The method does not change the original VLM architecture.

The experimental results and qualitative analysis show the effectiveness of the method.

**Weaknesses:**

The memory has a fixed size, so it becomes a bottleneck when processing long trajectories. In principle, it does not solve the problem that exists in the recurrent-based method.

The design of the compression module is similar to Q-Former, where the vision input is compressed to several fixed-sized vectors by cross-attention. The author needs to clarify the differences in the paper.

The core idea of using a small set of learnable queries to compress history has been applied in several existing VLM and LLM works, e.g., VidCompress, LVC, VoCo-LLaMA, etc. (just for example). What is the major difference between the proposed method and the existing compression methods, and why do you think they are not comparable?

The experiment is only conducted using Phi2+3B. It would be better to verify the effectiveness on different models.

**Questions:**

see in weaknesses

---

### Official Review · Reviewer_UFFT · 2025-11-03

**Soundness:** 3
**Presentation:** 3
**Contribution:** 3
**Rating:** 4
**Confidence:** 3

**Summary:**

This paper proposes a lightweight memory mechanism for vision-language models. Instead of concatenating past frames or using recurrent compression, the method dynamically compresses historical observations into a fixed-size memory bank and injects them across LLM layers via gated cross-attention.

This design preserves pretrained knowledge, reduces computational and memory overhead, and improves long-horizon reasoning. Experiments show efficiency gains and performance improvements over existing context-integration methods. The method is parameter-efficient, maintains linear complexity, and addresses temporal information retention in partially observable environments.

**Strengths:**

1. Practical and scalable approach for long-horizon multimodal reasoning.

2. Well-motivated architecture: decouples context compression and integration.

**Weaknesses:**

1. Since the compression relies on fixed learned queries, raising questions about whether it truly adapts to highly variable temporal structures versus performing a learned pooling.

2. The method does not benchmark against KV-cache compression / streaming attention / token eviction approaches, which are becoming standard for efficient long-context modeling in LLMs.

**Questions:**

1. How does the proposed dynamic compression differ in principle from learned token pooling or attention-based resampling? In what sense is the memory update truly adaptive to trajectory structure rather than fixed-query aggregation?

2. Can you comment on how DCA relates to KV-cache compression / streaming attention approaches in LLMs? Why were these not included as baselines, and do you expect them to be competitive or complementary in multimodal settings?

3. What is the wall-clock training overhead for the memory modules relative to standard finetuning?

4. In Table 3, the Navid baseline outperforms your method. Could you elaborate on why this occurs and explain whether the gap stems from architectural differences, training setups, or task–model alignment?

---

### Note · Authors · 2025-12-18

I have read and agree with the venue's withdrawal policy on behalf of myself and my co-authors.